# Beyond Therapeutic Adherence: Alternative Pathways for Understanding Medical Treatment in Type 1 Diabetes Mellitus

**DOI:** 10.3390/ijerph21030320

**Published:** 2024-03-09

**Authors:** Juan José Cleves-Valencia, Mónica Roncancio-Moreno, Raffaele De Luca Picione

**Affiliations:** 1Faculty of Psychology, Universidad del Valle, Calle 13 # 100-00, Cali 760042, Colombia; juan.cleves@correounivalle.edu.co (J.J.C.-V.); monica.roncancio@correounivalle.edu.co (M.R.-M.); 2Faculty of Law, Giustino Fortunato University, Via Delcogliano 12, 82100 Benevento, Italy

**Keywords:** adherence, diabetes mellitus type 1, semiotic cultural clinical psychology, narrative review

## Abstract

Given the psychosocial and economic costs of behaviors of patients who seem not to benefit from the medications, technologies, and medical therapies available for chronic diseases such as Type 1 Diabetes Mellitus, therapeutic adherence has been identified as one of the main focuses in the intervention. This paper presents contributions from semiotic cultural clinical psychology for understanding problems associated with the implementation of medical treatment in patients with Type 1 Diabetes Mellitus to explore psychological dimensions not yet reported in depth. A narrative review of 24 scientific articles published between 2012 and 2023 is carried out. The information is produced through thematic analysis, and the results are presented in three themes: 1. illness characteristics, 2. adherence and associated concepts, and 3. modes of intervention. It concludes with the development of a two-axis proposal for understanding the experience of patients that privileges psychological aspects involved in the disease and its treatment, considering the approach to the goals of treatment as dynamic and fluctuating rather than as final states.

## 1. Introduction

In the context of chronic diseases that require self-administered treatment, such as Type 1 Diabetes Mellitus (T1DM), therapeutic adherence has recently been identified as one of the main focuses of scientific research. Specifically, issues of patients who seem not to benefit from medications and medical therapies available in the field of health have been explored [1,2,3,4,5].

The empirical work reported has been abundant, with a predominance of the conception of adherence as a link that allows patients to achieve the expected biochemical results in medical care (i.e., in clinical laboratory tests). Assuming that adherence is fundamentally the degree to which a patient’s behavior corresponds to the prescriptions agreed upon with healthcare providers [6], much of the emphasis in many works lies in assessing patients’ behaviors and identifying associated factors.

This article, however, is guided by perspectives from Semiotic Cultural Clinical Psychology [7,8,9] to focus on the active processes of meaning-making about the illness experienced by patients. We conceive that this approach allows us to understand the changing and complex affective and cognitive dynamics that occur in what, for many, might seem simple and at the same time puzzling, namely, patients transitioning from the prescription of treatment to its implementation.

Similarly, T1DM is a chronic autoimmune disease in which the pancreas cannot produce insulin, and most patients with this diagnosis must be treated with exogenous and permanent supplies of insulin. In turn, developments in the health field have allowed the construction of various alternatives, such as the improvement of pharmacological therapies, the development of advanced therapies, the design and implementation of educational strategies, and the construction of technological devices for blood glucose monitoring [10,11].

In the medical care of the disease, health professionals pay special attention to biochemical indices of glycemic control that allow the evaluation of the closeness or distance with respect to treatment goals. Given the enormous challenge for patients to achieve these goals, lack of therapeutic adherence has been found to be the main cause of therapeutic failures and their associated problems, namely, intensification of treatments, increased hospitalizations, rising healthcare costs, and negative psychosocial implications for patients [12,13]. The situation has led different approaches from psychology and health sciences to report wide knowledge on the individual, singular, institutional, and social aspects involved in adherence [14,15,16,17,18]. Multiple actors, mediating variables, associated factors, and barriers have been identified. Evaluations of psychometric instruments have been developed, and protocols and intervention strategies of various kinds have been proposed [19,20].

All the advances and efforts that the issue of therapeutic adherence has managed to convene are a demonstration of the proportions that the concept has taken and point out the need to thoroughly elucidate its assumptions and possible implications for intervention since the problem shows no signs of stopping or improving. Hence, it is possible to identify a gap in existing knowledge. From our perspective, such a gap can begin to be bridged by conceptual tools that emphasize the patients’ suffering rather than the disease itself or the negative outcomes resulting from what some professionals may consider to be inadequate implementation of treatment. In the literature, there are some studies with similar interests in the meanings and experiences of patients. They are approaches from psychoanalysis or psychologies oriented via psychoanalytic theoretical developments that have contributed valuable reflections, in some cases referring directly to the concept of adherence [21,22,23]. Other antecedents can be found in works by the first author of this paper, oriented using cultural and psychoanalytic presuppositions [24,25], which have made it possible to begin to interrogate the scope of the concept of therapeutic adherence. According to these psychodynamic perspectives, we find that a series of conscious and unconscious, subjective, and relational processes are involved. This encompasses the development of personal identity, raising questions such as “Am I a sick person?” or “Do I have an illness?” It extends to considerations of autonomy and dependence, with individuals grappling with questions such as “Can I be responsible for my health?” and “Do I need others to deal with my condition?” Additionally, the degree of flexibility and rigidity in one’s conduct prompts reflections like “What can I do and what can’t I do?” and “How much can I deviate from prescribed rules and medical indications?” Emotional ambivalence further comes into play, exemplified by sentiments such as “I am grateful to those who support me” versus “I harbor deep anger and resentment towards myself and those close to me” [26].

The current proposal continues that impulse by relying on the semiotic approaches of cultural clinical psychology [27,28,29]. Thus, this paper aimed to approach adherence as a concept used for understanding problems associated with the implementation of medical treatment in patients with Type 1 Diabetes Mellitus (T1DM), with the purpose of making contributions that allow the elaboration of novel, enriched, and comprehensive interpretations. Specific objectives were established as follows:Explore the origin of the concept of therapeutic adherence to understand its evolution and context.Identify how therapeutic adherence is understood in the reviewed empirical studies with the purpose of contextualizing and understanding the different perspectives and approaches used in research.Explore the existing gaps in the concept of adherence to propose alternatives for exploring the sense-making processes surrounding the disease and its treatment.

Before presenting how the proposal was methodologically developed, some important features of the evolution of the concept of adherence will be outlined.

### Considerations on Adherence and Its Evolution

The concept of therapeutic adherence has undergone a complex and interesting evolution, of which some milestones will be presented. It should be noted that these milestones do not unfold in a linear fashion; rather, they are presented to suggest to the reader some implications for the treatment of patients diagnosed with T1DM. The concept began to take shape in the 1950s with a strong proximity to the idea of therapeutic compliance. It arose from the observation that patients’ behaviors in relation to the care of the disease were distant from the prescriptions of healthcare professionals.

In 1979, Haynes, Taylor, and Sackett proposed adherence as the degree of alignment between medical prescription and patient behavior [30]. While some authors find in their definition an attempt to distance from the idea of therapeutic compliance, others identify that it remains tied to a vertical relationship in which the autonomy that should be granted to patients regarding their treatment is not entirely clear [31]. In 1996, within the specific context of diabetes, researchers such as Hentinen and Kingäs postulated that adherence is not merely a passive process but an active and responsible one [32]. Their proposal highlights the importance of establishing a collaborative relationship between the patient and the healthcare personnel characterized by a more horizontal dynamic with possibilities for negotiation and active participation.

In 2001, the World Health Organization considered that a good starting point for defining adherence was to understand it as the extent to which the patient follows medical instructions. However, for the same institution, this definition had a couple of difficulties that may resemble the ones identified by Hentinen and Kingäs. Firstly, “following instructions” conveyed the idea of a subject as a receptacle awaiting the physician’s expert advice, and secondly, the definition did not recognize the full range of professions involved in the treatment of chronic diseases. For long-term therapies such as T1DM, adherence was defined two years later as the degree to which a person’s behavior—taking medication, following a diet, and/or making lifestyle changes—corresponded with the recommendations agreed upon with the health care provider. There is a shift in this definition, which now focuses on a broader spectrum than the doctor–patient binomial and proposes a type of relationship that allows patients to participate in their treatment because they “agree to”, instead of “following recommendations”, which are no longer “instructions”. However, some studies have identified that the difference between adherence and therapeutic compliance, which seems to have been conceptually resolved, can be blurred in the practice of patient care.

A review of the historical overview shows that the development of the concept of adherence involves different conceptions and is closely related to the notion of therapeutic compliance. Its common denominator, with certain nuances, is the request for patients to follow the medical prescription. With emphasis on the active meaning-making about the experience of becoming ill that can be derived from the approaches of Semiotic Cultural Clinical Psychology [33,34,35,36,37], we conceive that the notions employed by healthcare providers—of which therapeutic adherence is the prime example—require a close analysis as they guide interpretive frameworks and care practices in healthcare [38]. Therefore, we identify the need to approach the analytical tools used to tackle the difficulties presented by patients when trying to implement medical treatment, with the purpose of making contributions that allow the elaboration of novel, enriched, and comprehensive interpretations.

## 2. Materials and Methods

A narrative review was conducted, encompassing specialized literature in both English and Spanish. Among the expanding array of review methodologies [39], narrative review entails a descriptive synthesis of published literature, typically employing expert opinion-driven search strategies. This method offers a comprehensive overview of a specific topic, aiming to develop theoretical constructs, illuminate overlooked issues, or identify knowledge gaps to inform future research endeavors [40]. While inclusion and exclusion criteria are not mandatory in a narrative review, the present proposal did incorporate the development of such criteria for both article selection and literature retrieval (See Table 1. Search inclusion and exclusion criteria, Table 2. Selection inclusion and exclusion criteria, and Table 3. Criteria for the characterization of information).

Recognized thesauri within the scientific and international academic community were consulted. These were the DeCS (Health Sciences Descriptors) thesaurus, as an adapted and translated version of the MeSH (Medical Subject Headings) thesaurus. The search terms “Diabetes Mellitus Tipo 1” and “Adherencia al tratamiento”, along with their English equivalents “Type 1 Diabetes Mellitus” and “Treatment Adherence”, were employed using the Boolean operator “and.”

The search was carried out in the databases of the scientific and technical health information operator BIREME (Regional Library of Medicine for Latin America and the Caribbean). To obtain a global view of the phenomenon, articles in the databases Psy Articles, Fuente Académica Plus, Medline, Psychology and Behavioral Sciences Collection, and Gale Academic OneFile were reviewed with the same search terms and Boolean operator. The inclusion and exclusion criteria used are shown in Table 1 (Inclusion and exclusion criteria for the search).

Upon implementing the inclusion and exclusion criteria for the search, 63 articles were selected. After a general review of the abstract and structure of the articles, inclusion and exclusion criteria were constructed to establish and refine the studies that were analyzed. Table 2 (Inclusion and exclusion criteria for the selection) presents the four dimensions considered: 1. provided information, 2. study design, 3. study participants, and 4. type of publication. The implementation of the criteria led to the inclusion of 34 articles.

Subsequently, articles from scientific journals up to the third quartile (Q3) were included. This decision enabled the consideration of a diverse range of research relevant to analysis and appropriate for the research design. Article selection was based on the impact metrics of the consulted journals in Scimago, resulting in the final inclusion of a sample of 24 articles.

The analysis of the studies integrated two sequential procedures, and the methodological progression involved the establishment of differentiated units of analysis [41]. In the first procedure, the unit of analysis was the concept of adherence, and an initial reading of the introduction, methodology, and results sections in the articles was conducted. This reading was guided by semi-open categories constructed from the background, with descriptors developed from the reading to organize the information regarding the concept of adherence. The categories and their descriptors are presented in Table 3 (Criteria for the characterization of information).

Subsequently, all the information from the articles was examined in-depth through thematic analysis. This second procedure utilized patterns of meaning or “themes” in the data as the unit of analysis, allowing for the construction of a rich and detailed report consistent with the nature of a narrative review. The analysis was guided by grouping the information, constantly constructing and modifying the themes, and identifying similarities and differences in the dataset. Three themes were identified: 1. illness characteristics, 2. adherence and associated concepts, and 3. modes of intervention.

To ensure the integrity of the analysis process, three group meetings were conducted to discuss and reach agreements on procedural and analytical aspects that would guide the exercise. The principal researcher oversaw the analysis and held a pair of discussion meetings with one of the researchers. Subsequently, the process was validated by the third researcher. The analysis development process thus involved two phases of review and six adjustment sessions. Additionally, input was obtained from an external researcher, an expert in psychological research, who provided feedback on the manuscript’s methodological aspects through reading and commentary.

## 3. Results

The results are shown in two parts. On the one hand, Table 4 (Basic information of research included) shows the basic information on the included studies. On the other hand, the results of the analysis of the studies are presented in three themes: 1. illness characteristics, 2. adherence and concepts associated, and 3. modes of intervention.

### 3.1. Illness Characteristics

This theme indicates the features and particularities of the daily treatment and care of Type 1 Diabetes Mellitus (T1DM). It refers to therapeutic behaviors and transitions of patients in medical care, existing technologies for care, standards of glycemic control, and particularities of symptoms, as shown in Table 5 (Illness characteristics).

#### 3.1.1. Therapeutic Behaviors and Patient Transitions in Care

A chronic illness such as T1DM requires a considerable variety of therapeutic behaviors, such as counting carbohydrate, checking blood glucose multiple times a day, and administering insulin in response to blood glucose levels [42]. Some authors refer to treatment as a difficult, multi-step, complex, intensive, demanding, and highly visible regimen to others, which requires the adoption of a diet that keeps adequate glucose levels and the maintenance of physical activity [43,44,45,46].

Treatment also involves particularities at some key moments and transitions in the life cycle. The literature points out that it is necessary to assess educational needs and act accordingly at the reception of the diagnosis annually—according to the outcomes and emotional needs of the patients—when complications or factors impacting self-management arise and at the transition from one care system to another (i.e., from pediatric to adult care) [47]. In relation to life cycle transitions, studies report that the passage from childhood to adolescence is characterized as demanding for both patients and their families because it involves greater independence [48]. In turn, the transition to emerging adulthood, a period between the ages of 18 and 29 years, involves unique challenges such as the continuation in the exploration of identity, security in the world of work, and maintaining healthy social relationships. Finally, it is necessary to say that while some papers note that adolescents have the worst glycemic control averages, others indicate a concerning worsening between adolescence and adulthood [49].

#### 3.1.2. Existing Technologies and Standards for Glycemic Control

Existing technologies to support patients in the self-management of T1DM include devices, hardware, and software for glucose monitoring and adjustments to pharmacotherapy. Developments have included systems for automated insulin delivery using informed algorithms, in addition to more conventional options such as insulin delivered by syringes, pens or pumps, and glucose monitoring devices.

Some works find that the use of glucose monitoring devices allows better management of complications such as hypo- and hyperglycemia and better adherence to treatment, while some suggest that the use of continuous glucose monitoring integrated with continuous insulin infusion systems is associated with high levels of quality of life, fewer severe hypoglycemia, and fewer renal complications [50]. However, other studies report that, despite modern technological advances, many individuals experience high variability in glucose levels [51]. In this way, some research suggests that continuous glucose monitoring is probably more essential to achieving good glycemic control than the way insulin is administered (i.e., pen versus pump) [52].

Finally, some research mentions Time in Range (TIR) as a standard for reporting glycemic control [53]. Unlike HbA1c, as an index of average blood glucose over the last two to three months, TIR seeks to observe how long blood glucose levels may remain in or move away from a desirable range, as well as to identify patterns of adherence to medical treatment.

#### 3.1.3. Symptoms Particularities

Studies around T1DM symptoms point in various directions. Some point to the incidence of disease symptoms and their burden on quality of life understood as a construct with physical, psychological, and social dimensions of health [54]. Others highlight that symptoms of hypoglycemia and hyperglycemia make diabetes management a never-ending task that involves discomfort for the patient in terms of time, travel to places, and burden of care objects.

Hypoglycemia episodes have also been identified as barriers and challenges to adherence. Several patients report the difficulties in dealing with them and the cravings they experience with them. In addition to being situations in which the discomfort is more symptomatic, they generate significant social embarrassment, especially in adolescents and young adults. It is the most common and feared complication with the potential to condition the perception of severity, vulnerability, self-esteem, and self-efficacy.

### 3.2. Adherence and Associated Concepts 

The concept of therapeutic adherence is intertwined with diabetes management and diabetes outcomes. Some studies focus on exploring patients’ experiences in implementing medical treatment for the disease. Findings are presented below and summarized in Table 6 (Adherence and associated concepts).

#### 3.2.1. Adherence, Diabetes Management, and Diabetes Outcomes

Adherence appears with different connotations, whether as a means, a factor, or an outcome. Whether it is used dichotomously or as a continuum that allows observing degrees, the concept maintains an evaluative dimension of patients’ behavior regarding the implementation of medical prescriptions. In some studies, adherence is understood as a means to achieve clinical outcomes, and non-adherence is a barrier. Thus, it is positively associated with glycemic control and metabolic control [55,56,57].

In the works that define the concept of adherence, used globally (See Table 3. Criteria for the characterization of information), it is possible to identify that it is understood as the degree to which a person’s behavior corresponds with the advice of health care providers, close to that proposed by the World Health Organization [58].

Other approaches highlight the importance of daily consistency in adherence, indicating that it is composed of behavioral pillars such as self-monitoring of blood glucose, calculation of carbohydrate intake, and insulin administration. Around the idea of consistency in adherence begins to be seen that few patients follow the daily recommendations and that, although the achievement of the totality of them may be unrealistic, the recommendations are minimalistic in nature [59].

The concept of diabetes management, in turn, appears to encompass different dimensions than those of following medical prescriptions and is related to daily and adaptive strategies and actions by patients. Diabetes management involves interconnected aspects of self-regulation—understood as the management of one’s own emotions and thoughts (cognitions)—and social regulation—active use of relationships with others—to coordinate daily individual and social processes [60]. It is a concept that seems to be more effective when it occurs in the context of supportive and collaborative relationships and there are mediating effects, such as patient health communication and perceived treatment adherence, which can undermine it.

Diabetes outcomes, on the other hand, correspond to a notion whose emphasis lies on the effects or consequences of diabetes management or treatment adherence for patients. They are characterized by being measurable and can be subdivided into clinical, psychosocial, and behavioral outcomes. Clinical outcomes include biochemical indices such as HbA1c, preprandial and postprandial glucose, metabolic control, lipid and blood pressure profile, TIR, results of capillary glucometry measurements, index of metabolic control, and the patient’s lipid and blood pressure profile. Psychosocial and behavioral outcomes are usually measured by standardized instruments and include diabetes burnout, diabetes distress, psychosocial functioning at school, and communication skills [61].

#### General or Specific Uses of the Concept of Adherence

The concept of adherence is used globally (See Table 3. Criteria for the characterization of information) referring to self-care behaviors, to cornerstones such as self-monitoring of blood glucose, calculation of carbohydrate intake, and insulin administration, or to the totality of treatment behaviors [62]. In turn, it is specifically used in association with continuous glucose monitoring, diet, physical activity, and self-monitoring of blood glucose [63]. Regarding the general or specific uses of the concept, some works find that research tends to observe specific domains in adherence, so they seek to expand the existing literature by providing objective information obtained through measurement devices or recording of glucose behavior that allows inferring the adherence behaviors of patients from a general approach.

Adherence to the diet is understood as following the recommended guidelines, and it is observed that a mismatch between carbohydrate intake and insulin can result in hypo- or hyperglycemia. This highlights that a therapeutic behavior such as the diet involves an important series of analyses and considerations by patients.

#### 3.2.2. Studies Associated with Patients’ Experiences

Studies that emphasize the exploration of patient experience are characterized by highlighting psychosocial factors or by restituting meanings in patients’ process of understanding and caring for their disease. Some of them identify psychosocial factors that constitute barriers to improvements in self-care. These factors include emotional well-being, social support, self-efficacy, and personal motivation. Others suggest that it is necessary to understand the first year of diagnosis, such as the battle with symptoms, the emotions associated with the diagnosis, the challenges in the management of diabetes, and the positive outcomes recorded by patients [64]. In turn, through the development of a substantive theory, some studies find that at the center of the experience of giving meaning to the disease is the meaning of “becoming myself again”, which involves the reconstruction of interactions, emotions, and definitions of the disease, implying changes in the self of the patients.

### 3.3. Modes of Intervention

The main modes of intervention found are patient education—either through professionally led strategies with key thematic content or through experiential experiences—and shared medical appointments. They are presented below and summarized in Table 7 (Modes of intervention).

Some studies tackled the impact of participation in a structured therapeutic program administered by specialized nurses on both the perception of quality of life and treatment adherence among patients. This program encompassed instructions on insulin administration, self-monitoring of blood glucose levels, management strategies for hypoglycemia and hyperglycemia, dietary modifications tailored to individual patient needs, and recommendations for physical exercise [65].

Other studies implemented summer camps featuring diverse therapy modalities to assess their influence on Time in Range (TIR). These investigations involved rigorous glucose monitoring—conducted up to 18 times daily—and integrated multidisciplinary teams comprised of physicians, diabetologists, nurses, medical students, dietitians, and educators. Another similar approach investigated the effect of a four-session family-centered educational intervention. The sessions addressed physiological aspects of diabetes and its treatment, the importance of adherence, symptoms, and complications of the disease, and reasons for not accepting treatment.

On the other hand, shared medical appointments are reported as multicomponent interventions in multidisciplinary teams that focus on self-management, communication skills, goal setting, glucose pattern recognition, and peer/diabetes team support [50]. Finally, as potential sources of intervention, the consulted studies suggest considering the incidence of chronic stress associated with the demands of managing diabetes. It is also recommended to acknowledge the feelings of overwhelm, exhaustion, frustration, detachment, and helplessness commonly experienced in diabetes management. Furthermore, family stress emerges as a significant factor that may detrimentally affect patients’ treatment adherence and glycemic control.

## 4. Discussion

The diversity of research endeavors aimed at analyzing what happens with patients who do not seem to benefit from existing therapies and technologies for the treatment of Type 1 Diabetes Mellitus (T1DM) allows us to identify that the idea—in its basic essence—that patients need to follow the prescription of healthcare professionals may oversimplify the richness and range of patients’ experiences in implementing their medical treatments.

Even from the relatively limited conceptual scope of notions such as “diabetes management” and “diabetes outcomes”, cracks in therapeutic adherence begin to be observed as a concept that has gained significant proportions. These proportions can be appreciated in the abundant number of empirical works it has generated, and they contrast with the modest achievements pointed out by the broader dimensions that the issue increasingly takes on. The above suggests, in our view, that there are aspects that are not fully captured as the conceptual tools used emphasize the decisions, adaptations, and actions carried out by patients to take care of themselves or control the disease in its daily management. Similarly, the emphasis is also placed on the observable effects and consequences of poor management or poor adherence to the medical prescription (i.e., diabetes outcomes).

Indeed, the rigid and dichotomous conceptualizations of health and disease are unable to orient and handle the complexity of care in the case of chronicity. T1DM undermines any possibility of maintaining health and illness as two precise and well-defined notions. A processual vision is forced to deal with the question of what it means to support development and autonomization processes when we are in chronic conditions. How can we promote processes of progressive responsibility and reconstruction of future planning when we know that we will be ill for our whole lives? How can we promote the transition from positions of dependence on one’s medical conditions to more active and less delegating positions? How is it possible to develop a process of the subjective implication of the patient positioning, not only more in a position to adhere to medical prescriptions but also more aware of one’s own diagnosis, capable of acting correctly at the first signals and symptoms of the disease.

The approach based on the notion of therapeutic adherence—understood fundamentally as a follow-up or concordance of the patient’s behavior with the medical instruction—occupies a central place in the literature consulted. Adherence can become a readily available investigative tool due to its widespread use in empirical research. However, it has little conceptual development and shows a noticeable proximity to the notion of therapeutic compliance, from which it historically wanted to be detached as it attributed prescriptive and hierarchical traits. On the other hand, the scope of the concept also relates to how it can be perceived as an ideal by patients when used to refer to the entirety of therapeutic behaviors in treatment. Alternatively, opting to observe how long patients can maintain their glucose levels within the Time in Range (TIR) standards may overlook the high complexity of medical treatment, the experience of their symptoms, and the interplay of balances that patients must maintain in the disease and in other dimensions of their lives.

The studies of Cleves-Valencia and Ocampo-Cepeda enable us to understand the richness of tackling the meanings that patients continuously construct using comprehensive and flexible methodological alternatives. These studies have identified some scopes of the concept of therapeutic adherence, prompting researchers to analyze more clearly how it has been approached in the research. Now, from the Semiotic Cultural Clinical Psychology approach, it is possible to state that the approach to the patient’s relationship with the disease implies interpreting the questions that continually mobilize them to the continuous construction of meanings, as well as the permanent challenges they face in interpersonal and institutional relationships. This approach emphasizes the subjective and social aspects of the disease rather than focusing solely on therapeutic adherence, diabetes management, or its results.

One aspect referring to concerns about the sense of the self can be understood, for example, from the specific experience of a symptom such as hypoglycemia. The symptom brings into play an organic dimension (sweating, tremors, and alertness leading to excessive food consumption) that also implies the experience of a potentially threatened and contingent body. It can be interpreted that the patient’s life does not unfold in the silence of the organs as a metaphor that allows us to think about the partial loss of the capacity to maintain internal balance [66]. Silence is broken by the signs and symptoms of the disease and now presents to the patients the limitations of their bodies, questioning them about what they were, what they are, and what they may become. In other words, the disease and its symptoms have the potential to pose perplexing questions about the meaning of life and ways of living it.

We argue that it is difficult to understand these dimensions with a notion such as therapeutic adherence, which, in addition to the scopes that have been presented, tends to acquire an evaluative dimension. In practice, it can be used in a dichotomous way (adherent-non-adherent) or with qualifying adjectives (low, poor, good, and non-adherent). Another limitation of the notion of therapeutic adherence can be posed by the abundance of information provided by cases. An in-depth exploration of patients’ experiences allows us to understand that adherence to treatment is not identifiable with a feeling of personal satisfaction and that the quality of life that interests some studies includes more aspects than the decrease in episodes of crisis, the increase in TIR, and the decrease in hospitalizations. One of the cases addressed by Cleves–Valencia shows that adherence to treatment may be driven by reasons other than self-care, including risks to the body that may be unnoticed by the treating professionals due to the fascination with attaining therapeutic goals.

From a Semiotic Cultural Clinical Psychology approach, an emerging alternative is to understand patients’ decisions as carriers of meanings that remain, change, or may change. Through this perspective, it is possible to consider treatment goals as a constant search of patients, which need to be understood as resulting from the dynamic and permanent co-construction of meanings within the framework of changing events and interpersonal relationships. Seen from this perspective, the patients’ issues would not be read solely as errors in cognitive judgment or as matters related to the gradual appropriation of the disease linked to specific transition moments in the life cycle (childhood, adolescence, emerging adulthood, etc.), assuming that as a consequence of chronological time, there would be an end-point with positive results [67]. Rather, it might be considered that patients require psychological time to rediscover their bodies and the disease, and to reorientate in the world after the diagnosis. Therefore, we understand that at that moment, it is not enough for the patient to understand the disease in etiological and physiological terms, that is, to know what causes it, what its mechanisms are, and what the consequences of non-adherence to treatment are.

Although the Semiotic Cultural Clinical Psychology approach presented in this article has undergone several developments in recent years, it remains in a formative stage, necessitating further empirical investigations and the development of intervention models to elucidate its utility within psychology, particularly in clinical practice. We propose to consider that the experience of illness requires constantly dealing with uncertainty and unpredictability of the future and that patients may encounter conditions that exceed culturally shared interpretative systems, involving deep personal questionings about existence. Such an intensively effective position requires the development of processes of autonomy and development, subjective involvement, relational responsibility, and sharing of decision-making processes. Starting from the recognition of the traumatic-liminal experience of the illness, the keywords of a semiotic clinic research agenda are integration, flexibility, and contextualization [68]. The final purpose is not the realization of a perfect and nostalgic harmony between the doctor and the patient. Rather, the future aims to develop a gradual construction of competence and autonomy of all different subjects (we have indeed to consider that not only the patient is involved in the healthcare relationship) through the support, the confrontation, the dialogue, the semiotic translation of subjective meanings, and medical information in the relationship between physicians, patients, and families.

Thinking about people with T1DM as active individuals who constantly co-construct meanings transcends looking solely at behavior and static intrapsychic aspects. It seeks to overcome approaches that can be judgmental, evaluative, and reductionist of the complexity of patients’ experiences. We propose considering additional dimensions to those considered in conventional diabetes education. These involve identifying the specific forms, meanings, and moments in which the disease sufferer approaches or moves away from the goals of their treatment. Approaching the relationship of patients with the disease from a Semiotic Cultural Clinical Psychology perspective implies conceiving them as agents and recognizing that the unique aspects related to the gradual rediscovery of the body involve the elaboration of one’s own knowledge about the disease care. This discovery cannot be assured beforehand through universal prescriptions, but it can begin to be facilitated by conceptual notions that, rooted in academic culture, facilitate the transitions and discoveries of the patients.

## 5. Conclusions

With the ideas developed so far, we underline the need to understand the analytical tools and implications of the notions used in research and the applied field of health to identify what is requested from people with Type 1 Diabetes Mellitus (T1DM) when implementing medical treatment. We are interested in being attentive to the perspective held about patients, as we consider that it requires more than obedience and following the medical prescription on their part. Given the feelings of loss and confusion generated by the illness, we consider necessary to allow the co-construction of meanings about the illness and to facilitate the growth of autonomy and shared decision-making in a complex arena of interaction—among people, the disease, medical institutions, and their operators.

To contribute to the co-construction of such meanings by patients and based on the identification of the scope of available conceptual tools, we propose two axes that enable the exploration of patients’ experiences. These stem from the researchers’ psychotherapeutic experience and their idiographic approach to studying health. Specifically, they are grounded in the analysis of the discomfort experienced by individuals with T1DM and the obstacles they encounter when attempting to adhere to their treatment. They are proposed as guiding principles for exploring the meaning shaping patients’ experiences of illness. The included themes derive from the coding developed for the analysis of in-depth interviews conducted in a postgraduate research project, which implemented a grounded theory design. The axes do not aim to encompass the entire complexity and richness of the phenomenon and consider patients’ proximity to treatment goals as dynamic and fluctuating approaches.

**Axis** **1.**
*Identification of the ways, meanings, and moments in which the person with the illness approaches or moves away from the goals of their medical treatment in the context of attempting to implement the complex and varied therapeutic measures.*


To this end, it may be explored as follows:The presentation of the disease. This involves identifying the changes in the information about the disease that patients may present to others and the selection of the people to whom they make the diagnosis known.The moment of diagnosis. It involves understanding how the news was delivered, the circumstances surrounding it, the emotional reactions, the personal meanings that the diagnosis takes on, and the psychological implications of the immediate implementation of treatment.The alternatives for elaboration and healing. This consists of understanding the decision-making process carried out by the patient, from the reception of the diagnosis to finding a treatment or cure alternatives. It implies understanding their purpose and analyzing the discovery of possibilities or impossibilities of cure.The management of specific therapeutic behaviors. This involves understanding how the patients construct the knowledge or gradually appropriate the essential and differentiated therapeutic behaviors prescribed by medical expertise for the treatment of the illness, as well as the specific challenges that they entail.The sources of knowledge of the disease. It includes focusing on the analysis of the formal, informal, and experiential ways in which patients can elaborate an understanding of the illness and its behavior.The resulting sensation of implementing therapeutic behaviors. It implies considering the partial relinquishments of the pleasurable sensations and experiences that may be experienced when following the treatment and what this implies for beliefs about oneself while keeping the patient’s autonomy in mind.The purpose of implementing therapeutic behaviors. It involves understanding what drives patients to implement their treatment, beyond exclusively focusing on treatment goals.The relationship with health personnel and the treating institution. This encompasses understanding the interpersonal relationships among patients, professionals, and healthcare institutions, as they may either hinder the elaboration or expression of meanings in patients.

**Axis** **2.**
*Understanding the presence of the illness, its treatment, and other interpersonal relationships, ideals, and occupations in the daily life of patients.*


To this end, orientation can be provided as follows:The experience of organic symptoms. This involves exploring the bodily sensations implicated in the specific organic symptoms of T1DM, understanding what they may mean for the patient, and what existential questions they may raise for them.The continuous discovery of the body. It involves analyzing the re-signification of the body that patients may experience when trying to implement the treatment and experiencing the sensations involved in not meeting their demand for precision.Positioning towards the treatment. It involves being attentive to the ways in which patients can express their will regarding the available medical treatment alternatives.Valuation of the disease. It consists of appreciating the personal meanings that the disease, its care, and its effects acquire, how they vary, and within the framework of what interpersonal and institutional relationships such variations occur.Interests, beliefs, and occupations. This involves identifying the patient’s projection in terms of their life project and the ways in which human relationships, ideas, or activities can support their daily experience.

Contemporary healthcare scenarios are characterized by situations where chronic conditions are increasingly widespread and growing. This sociocultural frame is at risk of becoming transparent and taken for granted. The risk of stiffening and reduction of flexibility to signify one’s own experience is intrinsic to the condition of chronicity itself. Finally, an approach oriented by semiotic cultural clinical psychology invites us to foster dialogues between the general and anonymous categories of disease, health, and well-being (supported by the medical discourse and by the normative social framework) and the singular and contextual forms of sensemaking, without implying an impoverishment or a splitting of the subjective experience. Our critical reading of T1DM literature does not tend to find or just improve correct behaviors, pre-ordered protocols to follow, or strategic and efficient forms of persuasion for compliance. Rather, the semiotic development of one’s own illness experience can enhance the personal and affective value of these categorizations, their idiosyncratic aspects, the construction of new ways of relationships, and the progressive process of autonomy.

## Figures and Tables

**Table 1 ijerph-21-00320-t001:** Search inclusion and exclusion criteria.

Criteria	Inclusion	Exclusion
Time	2007–2023	Before 2007
Population	Type 1 Diabetes Mellitus	Type 2 Diabetes Mellitus Gestational Diabetes Monogenic diabetes syndromes, diseases of the exocrine pancreas, drug, or chemical induced diabetes
Language of publication	English and spanish	Different from english and spanish
Type of publication	Peer-reviewed publications	Scientific letters Book reviews Science outreach articles

**Table 2 ijerph-21-00320-t002:** Selection inclusion and exclusion criteria.

Criteria	Inclusion	Exclusion
Provided information	Empirical information: Case studies Group studies Population studies	Bibliographic information Secondary data analysis Psychometric instrument assessment Epidemiological studies
Study design	Quantitative Qualitative Mixed methods	Theoretical reflections Monographic articles Systematic reviews Narrative reviews State-of-the-art reviews Scoping review Meta-analysis Study protocols
Study participants	Type 1 Diabetes Mellitus	Mixed participants (DMT II y DMTI)
Type of publication	Peer reviewed papers	Academic conference presentations Academic event proceedings Abstract articles Serial publications Reference papers Article reviews Article series

**Table 3 ijerph-21-00320-t003:** Criteria for the characterization of information.

Analytical Categories	Descriptors	Explanation of Descriptors
Definition of adherence	Present	The article specifies what is understood by adherence or presents specific characteristics of the concept.
	Absent	It becomes necessary to infer what is meant by adherence in the article, based on its conceptual or methodological developments.
Approach to the concept of adherence *	Objective	In the data sources of the study, standardized instruments prevail, employing statistical or mathematical procedures to produce information.
	Subjective	In the data sources of the study, oral reports predominate, using hermeneutic procedures to produce information.
	Biochemical	In the data sources of the study, biochemical indices or technologies for blood glucose measurement prevail, employing statistical or mathematical procedures to produce information.
Type of use of the adherence concept	Specific use	The study refers to one or two therapeutic behaviors involved in the medical treatment of Type 1 Diabetes Mellitus.
	Global use	The study refers to three or more therapeutic behaviors involved in the medical treatment of Type 1 Diabetes Mellitus.

* This article takes as a reference point what is propounded by the World Health Organization (2003) regarding the study of adherence, but it suggests a modification in the understanding of “objective” and “subjective”. By “subjective”, it refers to the sources of information that allow a deeper comprehension of the meanings involved in indi-viduals’ experiences through idiographic procedures that uses oral reports.

**Table 4 ijerph-21-00320-t004:** Basic information of research included.

Authors	Year	Study	Journal	Quartile	Country	Participants
Gunns & Leach [42]	2020	An increased focus on stress for the management of blood glucose levels in type 1 diabetes: a case report	Australian Journal of Herbal and Naturopathic Medicine	Q3	Australia	1 Patient
Raymaekers, et al. [43]	2021	Diabetes-specific friend support in emerging adults with type 1 diabetes: Does satisfaction with support matter?	Journal of Behavioral Medicine	Q1	Belgium	324 Patients
Kelly, et al. [44]	2020	Adult attachment insecurity and associations with diabetes distress, daily stressful events and self-management in type 1 diabetes	Journal of Behavioral Medicine	Q 1	US	199 Patients
AlHaidar, et al. [45]	2020	Family Support and Its Association with Glycemic Control in Adolescents with Type 1 Diabetes Mellitus in Riyadh, Saudi Arabia	Journal of Diabetes Research	Q2	Saudi Arabia	56 Patients
AlBurno, et al. [46]	2022	Determinants of healthful eating and physical activity among adolescents and young adults with type 1 diabetes in Qatar: A qualitative study	PLoS ONE	Q1	Qatar	20 Patients
Vidal, et al. [47]	2022	Type 1 Diabetes Patient Experiences Before and After Transfer from a Paediatric to an Adult Hospital	Patient Preference and Adherence	Q1	Spain	11 Patients
Kim [48]	2022	Illness Experiences of Adolescents with Type 1 Diabetes	Journal of Diabetes Research	Q2	Korea	12 Patients
Floyd, et al. [49]	2016	Stabilization of glycemic control and improved quality of life using a shared medical appointment model in adolescents with type 1 diabetes in suboptimal control.	Pediatric Diabetes	Q1	US	32 Patients
Álvarez, et al. [50]	2021	Estudio de calidad de vida y adherencia al tratamiento en pacientes de 2 a 16 años con diabetes mellitus tipo 1 en Andalucía	Anales de Pediatría	Q3	Spain	178 Patients
Turton, et al. [51]	2023	Effects of a low-carbohydrate diet in adults with type 1 diabetes management: A single arm non-randomised clinical trial	PLoS ONE	Q1	Australia	16 Patients
Nagl, et al. [52]	2022	Time in Range in Children with Type 1 Diabetes before and during a Diabetes Camp—A Ceiling Effect?	Children (Switzerland)	Q2	Austria	26 Patients
Sladić Rimac, et al. [53]	2023	The Association of Personality Traits and Parameters of Glycemic Regulation in Type 1 Diabetes Mellitus Patients Using isCGM	Healthcare (Switzerland)	Q2	Croatia	155 Patients
Varni, et al. [54]	2018	Diabetes management mediating effects between diabetes symptoms and health-related quality of life in adolescents and young adults with type 1 diabetes	Pediatric Diabetes	Q1	US	418 Patients
Mahler, et al. [55]	2022	Perceived Family Stress Predicts Poor Metabolic Control in Pediatric Patients with Type 1 Diabetes: A Novel Triadic Approach	Journal of Diabetes Research	Q2	Switzerland	190 Patient- Caregiver triad
Baker, et al. [56]	2019	Structural model of patient-centered communication and diabetes management in early emerging adults at the transfer to adult care	Journal of Behavioral Medicine	Q1	US	247 Patients
Stoianova, et al. [57]	2018	Delay discounting associated with challenges to treatment adherence and glycemic control in young adults with type 1 diabetes	Behavioural Processes	Q2	US	267 Patients
Geneti, et al. [58]	2022	Adherence to Diabetes Self-Management and Its Associated Factors Among Adolescents Living with Type 1 Diabetes at Public Hospitals in Addis Ababa, Ethiopia: A Cross-Sectional Study	Diabetes, Metabolic Syndrome and Obesity: Targets and Therapy	Q2	Ethiopia	414 Patients
Westen, et al [59]	2018	Objectively measured adherence in adolescents with type 1 diabetes on multiple daily injections and insulin pump therapy.	Journal of Pediatric Psychology	Q1	US	80 Patients
Munion, et al. [60]	2020	The separation in coordination between social- and self-regulation for emerging adults with type 1 diabetes	Journal of Behavioral Medicine	Q1	US	212 Patients
Helgeson [61]	2021	Diabetes burnout among emerging adults with type 1 diabetes: a mixed methods investigation	Journal of Behavioral Medicine	Q1	US	88 Patients
Mansour, et al. [62]	2023	Does family-centred education improve treatment adherence, glycosylated haemoglobin and blood glucose level in patients with type 1 diabetes? A randomized clinical trial.	Nursing Open	Q1	Iran	60 Family members
Lv, et al. [63]	2021	Factors Associated with Adherence to Self-Monitoring of Blood Glucose Among Young People with Type 1 Diabetes in China: A Cross-Sectional Study	Patient Prefer Adherence	Q1	China	122 Patients
Nsamba, et al. [64]	2022	Lived Experiences of Newly Diagnosed Type 1 Diabetes Mellitus Children and Adolescents in Uganda	Journal of Multidisciplinary Healthcare	Q1	Uganda	20 Patients
Romero, et al. [65]	2022	Diabetes Management after a Therapeutic Education Program: A Qualitative Study	Healthcare (Switzerland)	Q2	Spain	18 Patients

**Table 5 ijerph-21-00320-t005:** Illness characteristics.

Theme 1. Illness Characteristics	Main Findings
	About therapeutic behaviors and patients transitions in care ✓ Illness requires a considerable variety of therapeutic behaviors (diet, exercise, and glucose monitoring) ✓ Treatment is referred to as difficult, multi-step, complex, and demanding ✓ Treatment involves particularities at some key moments and transitions in the life cycle About existing technologies and standards for glycemic control ✓ The use of glucose monitoring devices allows better management of complications ✓ Many individuals experience high variability in glucose levels despite modern technological advances ✓ Time in Range (TIR) is a standard for reporting glycemic control About symptoms particularities in T1DM ✓ The disease symptoms and their impact on quality of life ✓ Hypoglycemia and hyperglycemia make diabetes management a never-ending task ✓ Hypoglycemia episodes have the potential to condition the perception of severity, vulnerability, self-esteem, and self-efficacy

**Table 6 ijerph-21-00320-t006:** Adherence and associated concepts.

Theme 2: Adherence and Concepts Associated	Main Findings
	About adherence ✓ When defined, it is understood as the degree to which a person’s behavior corresponds with the advice of health care providers ✓ It has different connotations, whether as a means, a factor, or an outcome ✓ It is used dichotomously or as a continuum that allows observing degrees ✓ It is globally used referring to self-care behaviors, cornerstones, or the totality of treatment behaviors ✓ It is specifically used referring to continuous glucose monitoring, diet, physical activity, and self-monitoring of blood glucose About diabetes management and diabetes outcomes ✓ Diabetes management appears encompassing different dimensions than following medical prescriptions ✓ Diabetes management is related to daily and adaptive strategies and actions in treatment ✓ Diabetes outcomes focus on the effects of management or treatment adherence on patients ✓ Diabetes outcomes are characterized by being measurable About patient experiences ✓ Studies oriented towards patients’ experiences take into account psychosocial factors ✓ Patient experiences aims to restitute meanings in patients’ process of understanding and caring for their illness

**Table 7 ijerph-21-00320-t007:** Modes of intervention.

Theme 3. Modes of Intervention	Main Findings
	✓ Patient education through professionally led strategies with key thematic content ✓ Education through experiential strategies ✓ Family-centered educational strategies ✓ Multicomponent interventions in multidisciplinary teams

## Data Availability

Research data are available by writing to the author.

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
