# Peer review of "Beyond Therapeutic Adherence: Alternative Pathways for Understanding Medical Treatment in Type 1 Diabetes Mellitus"

_ijerph, 2024, doi:10.3390/ijerph21030320_

Round 1
Reviewer 1 Report
Comments and Suggestions for Authors
The authors made and relevant effort to propose a point of view from a theoretical approach, trying to provide enlightening from a different perspective about chronic patients.
I suggest that authors clarify the following topics:
1. On page 5, Results section Table 4, I think authors MUST use the PRISMA guide (http://www.prisma-statement.org/?AspxAutoDetectCookieSupport=1), at least to perform both, a basic qualitative and a quantitative evaluation of the included studies.
2. On this same Page 5, this table 4 could be improved if they added a column labeled "Level of the journal".
3. On page 8, The full 3.1 section would be improved if the authors summarized the text in a chart or graph. It doesn´t matter if that chart or graph is shown either descriptive or narrative.
4. On page 9, The full 3.2 section would be improved if the authors summarized the text in a chart or graph. It doesn´t matter if that chart or graph is shown either descriptive or narrative.
5. On page 11, The full 3.3 and 3.4 sections would be improved if the authors summarized the text in a chart or graph. It doesn´t matter if that chart or graph is shown either descriptive or narrative.
6. On pages 14 to 15, I propose to the authors the summarizing and reduction of the narrative in the discussion section, to emphasize clinical and PRACTICAL points, useful to clinicians. The named "Semiotic-Cultural Clinical Psychology" still requires experimental and basic clinical evidence, before being explored in patients. However, this study provides some hopeful and interesting findings.
7. On Page 16 (Axis 1 to 2), Please, authors should explain and show where in this paper these "Axes" were mentioned. Otherwise, please, authors, explain the origin of these "Axes".
Author Response
We are deeply grateful for your insightful comments, as we are confident, they enable us to enhance the quality of our work. We have meticulously reviewed and implemented them in accordance with the nature and purpose of the proposal we have developed. Your observations have provided a valuable opportunity to make additional adjustments to present the ideas and to refine the manuscript's wording accordingly. Throughout the manuscript, you will find sections highlighted in blue, indicating major modifications that have been made.
On page 5, Results section Table 4, I think authors MUST use the PRISMA guide (http://www.prisma-statement.org/?AspxAutoDetectCookieSupport=1), at least to perform both, a basic qualitative and a quantitative evaluation of the included studies:
In response to this helpful observation, objectives of the paper were developed to clear up its scope and rationale. The objectives were clarified as follows:
“This paper aimed to approach the adherence as a concept used for the understanding of problems associated with the implementation of medical treatment in patients with Type 1 Diabetes Mellitus (T1DM) with the pu-rpose of making contributions that allow the elaboration of novel, enriched and comprehensive interpretations. As specific objectives were stablished:
- Explore the origin of the concept of therapeutic adherence to understand its evolution and context.
- Identify how therapeutic adherence is understood in the reviewed empirical studies with the purpose of contextualizing and understanding the different perspectives and approaches used in research.
- Explore the existing gaps in the concept of adherence to propose alternatives for exploring the sense-making processes surrounding the disease and its treatment.”
Additionally, within the methodology section, a definition of a narrative review is provided, elucidating the distinctions from a systematic review or a scoping review:
“A narrative review was conducted, encompassing specialized literature in both English and Spanish. Among the expanding array of review methodologies (Grant & Booth, 2009), narrative review entails a descriptive synthesis of published literature, typically employing expert opinion-driven search strategies. This method offers a comprehensive overview of a specific topic, aiming to develop theoretical constructs, illuminate overlooked issues, or identify knowledge gaps to inform future research endeavors (Martín, et al., 2022). While inclusion and exclusion criteria are not mandatory in a narrative review, the present proposal did incorporate the development of such criteria for both literature retrieval and article selection”.
Information about the analytic procedure was also provided as follows:
“The analysis of the studies integrated two sequential procedures and the methodological progression involved the establishment of differentiated units of analysis. In the first procedure, the unit of analysis was the concept of adherence, and an initial reading of the introduction, methodology, and results sections in the articles was conducted. This reading was guided by semi-open categories constructed from the background, with descriptors developed from the reading to organize the information regarding the concept of adherence. The categories and their descriptors are presented in Table 3 (Criteria for the characterization of information).”
“Subsequently, all the information from the articles was examined in-depth through thematic analysis [41]. This second procedure utilized patterns of meaning or "themes" in the data as the unit of analysis, allowing for the construction of a rich and detailed report, consistent with the nature of a narrative review. The analysis was guided by grouping the information, constantly constructing, and modifying the themes, and identifying similari-ties and differences in the dataset. Three themes were identified, namely: 1. Illness characteristics, 2. Adherence and concepts associated, and 3. Modes of intervention.
“To ensure the integrity of the analysis process, three group meetings were conducted to discuss and reach agreements on procedural and analytical aspects that would guide the exercise. The principal researcher oversaw the analysis and held a pair of discussion meetings with one of the researchers. Subsequently, the process was validated by the third researcher. The analysis development process thus involved two phases of review and six adjustment sessions. Additionally, input was obtained from an external researcher, an expert in psychological research, who provided feedback on the manuscript's methodological aspects through reading and commentary”.
On this same Page 5, this table 4 could be improved if they added a column labeled "Level of the journal".
Following this higly important observation, we have appended a column to the table, providing information of the journals in which the articles were published, drawing from the information available in Scimago. In the methodology section, it is outlined:
“Upon implementing the inclusion and exclusion criteria for the search, 63 articles were selected. After a general review of the abstract and structure of the articles, inclusion and exclusion criteria were constructed to establish and refine the studies that were analyzed. Table 2 (Inclusion and exclusion criteria for the selection) presents the four dimensions considered, namely: 1. Provided information, 2. Study Design, 3. Study Participants, 4. Type of publication. The implementation of the criteria led to the inclusion of 34 articles.
Subsequently, articles from scientific journals up to the third quartile (Q3) were included. This decision enabled the consideration of a diverse range of research relevant for analysis and the nature of the design. Article selection was based on the impact metrics of the consulted journals in Scimago, resulting in the final inclusion of a sample of 25 articles.”
On page 8, The full 3.1 section would be improved if the authors summarized the text in a chart or graph. It doesn´t matter if that chart or graph is shown either descriptive or narrative.
This valuable observation enabled us to synthesize the findings and enhance their communicability. In the results section, a summary table (Table 5) was developed, facilitating readers' comprehension of main findings.
On page 9, The full 3.2 section would be improved if the authors summarized the text in a chart or graph. It doesn´t matter if that chart or graph is shown either descriptive or narrative
This valuable observation enabled us to synthesize the findings and improve their communicability. In the results section, a summary table (Table 6) was developed, facilitating readers' identification of main findings.
On page 11, The full 3.3 and 3.4 sections would be improved if the authors summarized the text in a chart or graph. It doesn´t matter if that chart or graph is shown either descriptive or narrative.
This valuable observation enabled us to synthesize the findings and enhance their communicability. Sections 3.3 and 3.4. were integrated based on their contribution to the research objectives and the characteristics of a narrative review. Similarly, in the results section, a summary table (Table 7) was developed, facilitating readers' access to the key findings.
On pages 14 to 15, I propose to the authors the summarizing and reduction of the narrative in the discussion section, to emphasize clinical and practical points.
This valuable feedback allowed us to synthesize the information presented in the discussion, in accordance with the objectives of the study; to identify key aspects and to recognize those that justify further exploration in future research.
The named "Semiotic-Cultural Clinical Psychology" still requires experimental and basic clinical evidence, before being explored in patients. However, this study provides some hopeful and interesting findings.
According with this interesting observation, a paragraph was included in the discussion as followed: “Although the Semiotic Cultural Clinical Psychology approach presented in this article has undergone several developments in recent years, it remains in a formative stage necessitating further empirical investigations and the development of intervention models to elucidate its utility within psychology, particularly in clinical practice.”
On Page 16 (Axis 1 to 2), Please, authors should explain and show where in this paper these "Axes" were mentioned. Otherwise,please, authors, explain the origin of these "Axes".
In response to this highly pertinent observation, we provide information regarding the origin of the axes and their intention, as follows:
“To contribute to the co-construction of such meanings by patients and based on the identification of the scope of available conceptual tools, we propose two axes that enable the exploration of patients' experiences. These stem from the researchers' psychotherapeutic experience and their idiographic approach to studying health. Specifically, they are grounded in the analysis of the discomfort experienced by individuals with Type 1 Diabetes Mellitus and the obstacles they encounter when attempting to adhere to their treatment. They are proposed as guiding principles for exploring the meanings shaping patients' experiences of illness and their relationship with treatment. The included themes derive from the coding developed for the analysis of in depth interviews conducted in a posgraduate research project interested in the psychological phenomena involved in T1DM, which implemented a grounded theory design. The axes do not aim to encompass the entire complexity and richness of the phenomenon and consider patients' proximity to treatment goals as dynamic and fluctuating approaches, rather than final states”.
Reviewer 2 Report
Comments and Suggestions for Authors
This paper explores the literature on adherence to type 1 diabetes treatments and proposes a new approach that ephasizes patient meaning making about the disease. I think the semiotic perspective is very interesting and the recommendation to focus on patient meanings is an important contribution to this field where blaming patients for willful non-adherence and over-emphasizing the cognitive aspects of adherence are often over-emphasized.
The paper can be significantly clearer about its goals: sometimes the narrative review seems to focus on how prior studies have conceptualized diabetes and adherence to treatment, while at other times it seems like a more traditional review of findings on the determinants of treatment adherence. A key requiremennt in a revision is to be much clearer about the goals of the review and the report about it.
The analytic approach is barely mentioned---I was not clear how the authors derived these four dimensions and what process they used to achieve inter-rater consistency in attributing points from prior studies to each.
The table describing the source papers could be much more informative if it also included info on definitions and findings that elucidate adherence that were drawn from each paper (paralleling the reporting of effect sizes in quantitatie reviews). It seemed like a few papers were cited more frequently than others and the reasons for this are not clear.
In the US and other countries, diabetes prevalence, treatment access/adherence differ by race/ethnicity, social class. health systems feature, but low income persons and persons from racial/ethnic excluded groups have lower access to appropriate care and lower adherence across multiple dimensions. It seems to me that any new model of adherence needs to talk about biases in care and the relative quality of care for low income/publically financed patients, and the role of financial costs in accessing medications, healthier food, recreational opportunities etc.
Sections 3.4.1 and 3.4.2 seem like lists of random additional findings and don't seem to add much to the analysis as presented.
I really did not understand the articulation of 2 axes in the conclusion. Are these proposed as ways of thinking about diabetes T1 adherence, as sources for alternative approaches to analyzing treatment failure in T1, as bases for qualitative assesment or development of interventions, or something else? And the two "axes" seem like collections of themes, concept buckets, but not as coherent dimensions or variables.....Perhaps the authors could say more about what they mean by "axis" and how items are added to an axis, and why the themes attributed to each are joined together, and how this new model would be applied in research and practice.
Comments on the Quality of English LanguageEnglish is excellent, but the writing not so much. There are long sentences that are hard to follow and paragraphs seem to address disparate topics without transitions.
Author Response
We are deeply grateful for your insightful comments, as we are confident they enable us to enhance the quality of our work. We have meticulously reviewed and implemented them in accordance with the nature and purpose of the proposal we have developed. Your observations have provided a valuable opportunity to make additional adjustments to present the ideas and to refine the manuscript's wording accordingly. Throughout the manuscript, you will find sections highlighted in blue, indicating major modifications that have been made to the manuscript.
Sometimes the narrative review seems to focus on how prior studies have conceptualized diabetes and adherence to treatment, while at other times it seems like a more traditional review of findings on the determinants of treatment adherence. A key requirement in a revision is to be much clearer about the goals of the review.
In response to this helpful observation, objectives of the paper were developed in order to clear up its scope and rationale. The objectives were clarified as follows:
“This paper aimed to approach the adherence as a concept used for the understanding of problems associated with the implementation of medical treatment in patients with Type 1 Diabetes Mellitus (T1DM) with the pu-rpose of making contributions that allow the elaboration of novel, enriched and comprehensive interpretations. As specific objectives were stablished:
- Explore the origin of the concept of therapeutic adherence to understand its evolution and context.
- Identify how therapeutic adherence is understood in the reviewed empirical studies with the purpose of contextualizing and understanding the different perspectives and approaches used in research.
- Explore the existing gaps in the concept of adherence to propose alternatives for exploring the sense-making processes surrounding the disease and its treatment.
The analytic approach is barely mentioned---I was not clear how the authors derived these four dimensions and what process they used to achieve inter-rater consistency in attributing points from prior studies to each.
This important observation allowed us to refine the analytical approach in the subsequent paragraphs:
“The analysis of the studies integrated two sequential procedures and the methodological progression involved the establishment of differentiated units of analysis. In the first procedure, the unit of analysis was the concept of adherence, and an initial reading of the introduction, methodology, and results sections in the articles was conducted. This reading was guided by semi-open categories constructed from the background, with descriptors developed from the reading to organize the information regarding the concept of adherence. The categories and their descriptors are presented in Table 3 (Criteria for the characterization of information).
Subsequently, all the information from the articles was examined in-depth through thematic analysis. This second procedure utilized patterns of meaning or "themes" in the data as the unit of analysis, allowing for the construction of a rich and detailed report, consistent with the nature of a narrative review. The analysis was guided by grouping the information, constantly constructing, and modifying the themes, and identifying similarities and differences in the dataset. Three themes were identified, namely: 1. Characteristics of the disease, 2. Concepts associated with the patient's relationship with treatment, and 3. Modes of intervention.
To ensure the integrity of the analysis process, three group meetings were conducted to discuss and reach agreements on procedural and analytical aspects that would guide the exercise. The principal researcher oversaw the analysis and held a pair of discussion meetings with one of the researchers. Subsequently, the process was validated by the third researcher. The analysis development process thus involved two phases of review and six adjustment sessions. Additionally, input was obtained from an external researcher, an expert in psychological research, who provided feedback on the manuscript's methodological aspects through reading and commentary.
The table describing the source papers could be much more informative if it also included info on definitions and findings that elucidate adherence that were drawn from each paper (paralleling the reporting of effect sizes in quantitative reviews)
In the methodology section, a definition of what a narrative review entails is provided, enabling readers to understand why it differs from a systematic review or a scoping review (both of which requires paralleling the reporting of effect sizes):
A narrative review was conducted, encompassing specialized literature in both English and Spanish. Among the expanding array of review methodologies (Grant & Booth, 2009), narrative review entails a descriptive synthesis of published literature, typically employing expert opinion-driven search strategies. This method offers a comprehensive overview of a specific topic, aiming to develop theoretical constructs, illuminate overlooked issues, or identify knowledge gaps to inform future research endeavors (Martín, et al., 2022). While inclusion and exclusion criteria are not mandatory in a narrative review, the present proposal did incorporate the development of such criteria for both literature retrieval and article selection.
It seemed like a few papers were cited more frequently than others and the reasons for this are not clear.
While this valuable observation enabled us to review all citations within the document and make corresponding adjustments to ensure equitable representation of works, the thematic analysis conducted permits the development of non-exclusively categorizable themes. Thus, it is pertinent to cite certain works more than once to underscore their relevance across various thematic dimensions. This approach ensures a comprehensive representation of the literature and facilitates a nuanced discussion of the multifaceted aspects addressed by the cited studies.
It seems to me that any new model of adherence needs to talk about biases in care and the relative quality of care for low income/publically financed patients, and the role of financial costs in accessing medications, healthier food, recreational opportunities etc.
Although this remark is interesting, it exceed our scope. Objectives of the paper were developed in order to clear up its scope and rationale. The objectives were clarified as follows:
“This paper aimed to approach the adherence as a concept used for the understanding of problems associated with the implementation of medical treatment in patients with Type 1 Diabetes Mellitus (T1DM) with the pu-rpose of making contributions that allow the elaboration of novel, enriched and comprehensive interpretations. As specific objectives were stablished:
- Explore the origin of the concept of therapeutic adherence to understand its evolution and context.
- Identify how therapeutic adherence is understood in the reviewed empirical studies with the purpose of contextualizing and understanding the different perspectives and approaches used in research.
- Explore the existing gaps in the concept of adherence to propose alternatives for exploring the sense-making processes surrounding the disease and its treatment.”
Sections 3.4.1 and 3.4.2 seem like lists of random additional findings and don't seem to add much to the analysis as presented.
This valuable observation enabled us to synthesize the findings and enhance their communicability. Sections 3.4.1 and 3.4.2 were integrated based on their contribution to the research objectives and the characteristics of a narrative review. Similarly, in the results section, a summary table (Tables 5, 6, and 7) was developed for each theme derived from the thematic analysis, facilitating readers' access to the key findings.
The authors could say more about what they mean by "axis" and how items are added to an axis, and why the themes attributed to each are joined together,and how this new model would be applied in research and practice.
In response to this highly pertinent observation, we provide information regarding the origin of the axes and their intention, as follows:
“To contribute to the co-construction of such meanings by patients and based on the identification of the scope of available conceptual tools, we propose two axes that enable the exploration of patients' experiences. These stem from the researchers' psychotherapeutic experience and their idiographic approach to studying health. Specifically, they are grounded in the analysis of the discomfort experienced by individuals with Type 1 Diabetes Mellitus and the obstacles they encounter when attempting to adhere to their treatment. They are proposed as guiding principles for exploring the meanings shaping patients' experiences of illness and their relationship with treatment. The included themes derive from the coding developed for the analysis of in depth interviews conducted in a posgraduate research project interested in the psychological phenomena involved in T1DM, which implemented a grounded theory design. The axes do not aim to encompass the entire complexity and richness of the phenomenon and consider patients' proximity to treatment goals as dynamic and fluctuating approaches, rather than final states”.
Round 2
Reviewer 1 Report
Comments and Suggestions for Authors
Thanks for your answers, your team did an excellent job by making strong changes and specifications to the article.
Reviewer 2 Report
Comments and Suggestions for Authors
The paper should be accepted as here. The current revisions respond to reviewer comments and the paper is well presented.
Comments on the Quality of English Languagepaper is in good shape. I suggest one more read through to check punctuation in the new sections and tables.